# Differences between the Leaf Mycobiome of *Coffea arabica* and Wild Coffee Species and Their Modulation by Caffeine/Chlorogenic Acid Content

**DOI:** 10.3390/microorganisms9112296

**Published:** 2021-11-05

**Authors:** Leandro Pio de Sousa, Oliveiro Guerreiro Filho, Jorge Maurício Costa Mondego

**Affiliations:** 1Department of Genetic, Evolution, Microbiology and Immunology, Institute of Biology, State University of Campinas, Campinas 13020-902, Brazil; 2Centro de Café ‘Alcides Carvalho’, Instituto Agronômico (IAC), Campinas 13020-902, Brazil; oliveiro.guerreiro@sp.gov.br; 3Centro de Pesquisa e Desenvolvimento de Recursos Genéticos Vegetais, Instituto Agronômico (IAC), Campinas 13020-902, Brazil; jmcmondego@gmail.com

**Keywords:** mycobiome, caffeine, chlorogenic acid, *Coffea* sp.

## Abstract

The study of microbes associated with the coffee tree has been gaining strength in recent years. In this work, we compared the leaf mycobiome of the traditional crop *Coffea arabica* with wild species *Coffea racemosa* and *Coffea stenophylla* using ITS sequencing for qualitative information and real-time PCR for quantitative information, seeking to relate the mycobiomes with the content of caffeine and chlorogenic acid in leaves. *Dothideomycetes*, *Wallemiomycetes*, and *Tremellomycetes* are the dominant classes of fungi. The core leaf mycobiome among the three *Coffea* species is formed by *Hannaella*, *Cladosporium*, *Cryptococcus*, *Erythrobasidium,* and *Alternaria*. A network analysis showed that *Phoma*, an important *C. arabica* pathogen, is negatively related to six fungal species present in *C. racemosa* and *C. stenophylla* and absent in *C. arabica*. Finally, *C. arabica* have more than 35 times the concentration of caffeine and 2.5 times the concentration of chlorogenic acid than *C. stenophylla* and *C. racemosa*. The relationship between caffeine/chlorogenic acid content, the leaf mycobiome, and genotype pathogen resistance is discussed.

## 1. Introduction

The study of the leaf microbiome has gained a lot of interest because it has been widely reported that these microbes can protect plants against pathogens [1], and, therefore, this ability could be used to avoid environmentally harmful pest/disease control strategies. Despite this practical interest, there is still a lot to understand about the ecology of the leaf environment, which is called the phyllosphere. An important issue concerning the components that shape the phyllospheric community is that different varieties of the same species can support different communities [2], as different developmental stages of the same plant exhibit community disparities [3]. The dynamics between these biotic factors and environmental factors must be better understood to investigate a possible practical application.

*Coffea* (Rubiaceae) is a genus with 130 species from tropical Africa, Madagascar, and the Mascarene Islands [4]. Among these species, *C. arabica* and *C. canephora* are the most cultivated for the production of beverages, making coffee one of the most exported commodities in the world [5], being a source of subsistence of more than 125 million people in Latin America, Africa, and Asia [6]. *Coffea arabica* is endemic to Ethiopia, Sudan, and Northern Kenya, while *C. canephora* is found in various countries throughout tropical Africa [4]. Other species such as *C. racemosa,* found in East Africa, and *Coffea stenophylla,* found in Guinea, Ivory Coast, Liberia, and Sierra Leone, are also planted, but on a smaller scale, not having great economic importance [7]. These species have interesting characteristics, such as greater resistance to diseases and abiotic stresses [8].

The study of microbes associated with the coffee tree is still in its infancy compared to other species [9,10,11], but it has been gaining attention in recent years [12,13,14,15], predominantly in *C. arabica*. This interest is based on two targets: (a) the discovery of microbes that can be used for biocontrol of diseases and pests; (b) to understand the influence of the host on the microbiome. Specifically, in the case of fungi, Santamaría and Bayman 2005 [9] isolated the genera *Pestalotia* and *Botryosphaeria* from the leaves of *C. arabica* planted in Puerto Rico. Saucedo-García et al. (2014) and Bongiorno et al. (2016) [12,13] working in Mexico and Brazil with endophytic fungi, detected the predominance of *Colletotrichum*, and *Xylaria* in Mexico, and *Colletotrichum*, *Trichoderma, Schizophyllum, Mycosphaerella, Cladosporium,* and, *Cercospora* in Brazil, respectively, suggesting that the mycobiota varies according to the region where the coffee tree grows.

In this work, we compared the leaf mycobiome (fungal microbiome) of the traditional crop *C. arabica* with *C. racemosa* and *C. stenophylla* using ITS (intergenic transcribed spacer) sequencing for qualitative information and real-time PCR for quantitative information, seeking to relate secondary metabolites involved in plant defense against pathogens to the content of caffeine and chlorogenic acid in leaves.

## 2. Methods

### 2.1. Biological Material

Adult coffee plants stored in the germplasm bank of the Instituto Agronômico de Campinas, Sao Paulo, Brazil (22°53′ S 47°5′ W, 664 m a.s.l.), were used for this study. The samples were taken on 28 July 2021 at 10 a.m. and 20% relative humidity. The Brazilian southeastern dry season was chosen in order to avoid washing-off of the phyllosphere, which could have interfered with diversity analysis. *C. arabica* cv Catuaí Vermelho IAC 81, *C. stenophylla* IAC 1090, and *C. racemosa* IAC 1193-3-2 were the plants chosen for this study. Leaves were collected from three individuals of each cultivar. Five mature leaves were detached from each plant on the second pair of leaves of each branch. Leaves were about 1.2 m from the ground and on different branches without traces of damage from pests and diseases. The samples were collected with a glove, placed in appropriate paper bags, and immediately taken to be analyzed.

### 2.2. Microbes Collection, DNA Extraction, and Sequencing

Fungi cells on the surface of leaves were sampled with sterilized cotton-tipped swabs moistened with 0.9% NaCl solution. DNA extraction was carried out with magnetic beads (MagMax^®^ ThermoFisher Scientific, Waltham, MA, USA) according to manufacturer’s protocol. The identification of fungi was performed using high-performance sequencing of the ITS1 region. Amplification was performed with ITS1 (GAACCWGCGGARGGATCA [16]) and ITS2 (GCTGCGTTCTTCATCGATGC [17]) primers. The PCR reactions were carried out using Platinum Taq (Invitrogen, Waltham, MA, USA). The PCR products were purified using the QIAquick Gel Extraction Kit (Qiagen, Hilden, Germany). Libraries were prepared using TruSeq DNA Sample Prep Kits (Illumina, San Diego, CA, USA) and sequenced in a MiSeq system using the standard Illumina primers provided by the manufacturer. A single-end 300 nucleotide run was performed. The sequences were deposited on BioProject PRJNA760679.

### 2.3. Pre-sequence Analysis

All sequences generated were processed using QIIME2, according to Vernier et al. [18]. High-quality sequences (>200 bp in length, quality score > 25 according to QIIME parameters) were trimmed and clustered into operational taxonomic units (OTUs) at 97% sequence identity using Mothur 1.44.3 (https://github.com/mothur/mothur/releases, accessed on 21 September 2021). Representative sequences for each OTU were then aligned using PyNAST and assigned taxonomy with the RDP classifier (http://rdp.cme.msu.edu, accessed on 21 September 2021), according to Flores et al. [19].

### 2.4. Identification of Differentially Abundant Taxonomic Groups

Differences between abundances of taxonomic groups were visualized through non-metric multidimensional scaling (NMDS) plots using Bray–Curtis dissimilarity. NMDS plots were generated using the discriminatory genera between the cohorts. Vegan and ggplot2 packages for R were used for these purposes.

### 2.5. Interaction Networks between Mycobiomes

Spearman correlation coefficient (r), which shows pairwise correlations between abundances of different fungal genera, was used to create a separate interaction network, using the tool CoNet in Cytoscape 3.7.1. Coefficient values greater than or equal to 0.6 with a *p*-value greater than 0.05 were transformed into links between two genera in the genus network.

### 2.6. Estimation of Fungal Abundance

For a more quantitative description, StepOne™ Real-Time PCR System (Applied Biosystems^®^, Foster City, CA, USA) was used to estimate total fungal abundance in the samples. The primers FR1 (5′-AICCATTCAATCGGTAIT-3′)/FF390 (5′-CGATAACGAACGAGACCT-3′) for 18S rRNA gene were used according to the specifications of Chemidlin et al. [20]. The concentration of the total DNA (ng/µL) was determined using Qubit fluorometry (Invitrogen™, Carlsbad, CA, USA). Real-time PCR products obtained from DNA from a pure culture of *Trichoderma harzianum* were cloned in a plasmid (pGEM-T Easy Vector System, Promega, Madison, WI, USA) and used as standard. The measurements were carried out in triplicates.

### 2.7. Caffeine and Chlorogenic Acid Analysis

The same leaves used previously for DNA extraction were dried at 80 °C and ground to a powder with a mortar and pestle for further extraction and analysis. Caffeine (CAF) and chlorogenic acids (CGA) were extracted with 80% methanol at 60 °C and 70% ethanol at 50 °C, respectively, and analyzed by high-performance liquid chromatography (Shimadzu LC-10 liquid chromatography system, Sao Paulo, Brazil) on a Ultropac TSK ODS-120T column (Sigma-Aldrich, San Luis, MO, USA), according to the specifications of Mazzafera et al. [21] and Ramiro et al. [22]. The mobile phase for the elution of CGA and CAF was a mixture of acetonitrile and 0.2M H_3_PO_4_ (11:89 *v*/*v*). The mobile phase was filtered through a 0.20 μm microfilter. The total runtime of CGA and CAF was 15 min, with a flow rate of 0.23 mL/min and column temperature of 50 °C. CGA and CAF were monitored between 190 and 400 nm. Pure caffeine (Sigma-Aldrich, San Luis, MO, USA) and chlorogenic acid (Sigma-Aldrich, San Luis, MO, USA) were used as standard, and values were expressed as milligrams of compound per gram of leaves. Each sample was extracted and analyzed three times with similar results.

### 2.8. Statistical Analysis

Differences between fungal abundance were tested for significance with a Tukey test at a *p*-value of <0.05. Analysis of similarity (ANOSIM) was conducted to assess relative similarity of fungal communities. All statistical analyses were performed in the R package.

## 3. Results

### 3.1. Fungal Abundance

For a more quantitative description of fungal abundance, real-time PCR was performed, which detects the number of copies per ng of DNA (Figure 1a). The experiment showed that the fungal population was highly variable and statistically different between *Coffea* species (Figure 1a). *C. stenophylla* showed a wider population, while *C. arabica* showed a narrower population.

### 3.2. Fungal Species Composition

The Ascomycota and Basidiomycota were the phyla found in our samples, the dominant classes being *Dothideomycetes*, *Wallemiomycetes*, and *Tremellomycetes* (Figure 1b). The core mycobiome (fungi found in the three coffee trees) is formed by *Hannaella*, *Cryptococcus*, *Erythrobasidium, Cladosporium,* and *Alternaria* (Figure 2).

### 3.3. Differentially Abundant Fungal Genera in the Phyllosphere

Beta diversity analysis using NMDS plots (Figure 3) based on Bray–Curtis dissimilarity of discriminating genera shows that the mycobiomes slightly segregate (*p*-value = 0.05495), although they are significantly different (P_ANOSIM_ < 0.05).

### 3.4. Correlation and Network Analysis

For the correlation analysis (Spearman correlation), genera with abundance >0.5% were used. Figure 4 shows seven genera with a strong negative correlation with *Phoma* (blue thicker lines) and nine genera with a strong positive correlation with *Botrytis* (red thicker lines).

### 3.5. Caffeine and Chlorogenic Acids

The concentrations of caffeine and chlorogenic acid were measured by high-performance liquid chromatography (HPLC). *C. arabica* had higher concentrations of CAF/CGA than those of *C. racemosa* and *C. stenophylla* (Table 1).

## 4. Discussion

In this study, we analyzed the phyllosphere of three different coffee trees, one traditionally cultivated (*C arabica*) and two with potential for cultivation (*C. racemosa* and *C. stenophylla*). The connection between caffeine/chlorogenic acid content, the leaf mycobiome, and genotype resistance against pathogens is discussed.

We investigated the concentration of caffeine (CAF) and chlorogenic acid (CGA) in coffee leaves. Both are secondary metabolites produced by various plants and are important in the plant’s ability to defend against pathogens and competing plants [23,24]. In *Coffea*, CAF and CGA have been suggested as important to suppress the growth of pathogens and also to favor antagonists of these pathogens [25,26]. For example, in soil, CAF enhanced the mycoparasitism of *Trichoderma* up to 1.7-fold [25]. Moreover, CGA inhibits the spore germination and mycelial growth of *Sclerotinia, Fusarium, Botrytis,* and *Cercospora* [26]. Thus, CAF and CGA select microorganisms that tolerate higher concentrations of these metabolites. Furthermore, these microbes can even use CAF/CGA as sources of carbon and nitrogen [27,28]. *C. arabica* have more than 35 times the concentration of CAF and 2.5 times the concentration of CGA than *C. stenophylla* and *C. racemosa*. This characteristic of *C. arabica* was inherited from the parental *C. canephora* [29]. Yadav et al. [30] showed that the bacterial community in the phyllosphere of Mediterranean plants is also shaped by the chemical content of the leaves, since the size of the bacterial community is negatively correlated with total phenolic content. Interestingly, the fungi population found in *C. arabica*, which was inferred by real-time PCR, proved to be much lower than that in *C. stenophylla* and *C. racemosa* (Figure 1a). This suggests that the concentration of CAF/CGA in *C. arabica* may select tolerant fungi to inhabit the leaves. We propose that the high concentration of CAF/CGA in *C. arabica* may have helped to shape the structure of the fungi community. Following this hypothesis, as *C. racemosa* and *C. stenophylla* produce low concentrations of CAF/CGA, the mycobiome is wider (Figure 2).

Another possibility for the less abundant mycobiome of *C. arabica* could be the previously reported phenomenon that domesticated plants learn to have a loss of diversity in their associated microbes compared to wild plants [31]. In the case of *C. arabica*, the high concentration of CAF/CGA and the domestication process could both helped to shape the mycobiome. To what degree each phenomenon (high CAF/CGA and domestication) contributes to shaping the mycobiomes should be further investigated.

NMDS data (Figure 3) reflect that the *C. arabica* fungal community appears to be more related to the *C. stenophylla* community. In fact, *C. arabica* and *C. stenophylla* are phylogenetically more closely related than *C. racemosa* [30], so the fungi communities of *C. arabica* and *C. stenophylla* were expected to be more similar than that of *C. racemosa*.

Another aspect to be discussed is the structure of the fungal community. The prevalent fungi are yeasts such as *Hannaella*, *Aureobasidium*, *Erythrobasidium,* and *Papiliotrema*, but molds such as *Cladosporium* were also found in abundance (Figure 2). The core mycobiome is quite restricted with five genera (*Hannaella*, *Cryptococcus*, *Erythrobasidium, Cladosporium,* and *Alternaria*; Figure 2) commonly found in soil and in association with plants. Several plant pathogens have been identified in our samples, namely, *Nigrospora*, *Botrytis*, *Pseudocercospora*, *Pallidocercospora*, *Passarola* (found only in *C. stenophylla*), *Fusarium* (found in *C. stenophylla* and *C. racemosa*), *Phoma* (found only in *C. arabica*), and *Alternaria* (found in all three *Coffea*). Of these, only *Phoma* and *Fusarium* are known to cause coffee diseases. *Phoma* causes Phoma leaf spot [32], and *Fusarium* is the causative agent of coffee wilt [33]. Other important pathogens for coffee, such as *Cercospora* and *Hemileia,* were not detected. The correlation and network analysis showed that *Phoma* is strongly negatively influenced by eight genera (Figure 4), six of which were not found in *C. arabica*. It is known that *C. arabica*, because of its smaller genetic base and recent speciation, is more susceptible to pathogens, despite the high CAF/CGA production [34]. The absence of most fungi in *C. arabica* (including *Phoma* antagonists) indicates that the complex interaction network of the various *C. racemosa/C. stenophylla* fungi may protect them from pathogens, which is seen in practice, because non-domesticated coffee plants are more resistant to diseases [35]. In fact, higher microbiome diversity is known to provide several benefits to the host [36]. There is evidence of the correlation between microbial diversity and resistance to competitors, where the wider and more diverse the microbial community, the more niches occupied, leading to a decrease in opportunities for invasive microbes [36]. Therefore, we propose that the poor *C. arabica* mycobiome (side effect of the domestication process and high CAF/CGA production) may impact its susceptibility to pathogens such as *Phoma*. However, this relationship appears to be more complex. Although *C. arabica* produces more CAF/CGA, it is more susceptible to pathogens, while *C. racemosa* and *C. stenophylla* produce less CAF/CGA and are less susceptible. Among the cultivars of *C. arabica*, those that produce more CAF/CGA are less susceptible to pathogens [37], showing that the production of these compounds is, in fact, linked to pathogen resistance.

Therefore, to accommodate the various conflicting information, we propose the following scenario: the high concentration of CAF/CGA in *C. arabica*, on the one hand, tries to inhibit possible pathogens; on the other hand, it decreases the leaf mycobiome abundance, allowing opportunistic (i.e., *Phoma*) microbes to install themselves more easily. *C. racemosa* and *C. stenophylla* produce less CAF/CGA and harbor a more complex mycobiome that protects them from more severe infections. An approach to testing this hypothesis could be to manipulate the fungal diversity of *C. arabica*, testing whether the severity decreases or not. Several fungi that antagonize with *Phoma* are not found in *C. arabica*. It may be interesting to test the effect of inoculation of these putative antagonists in *C. arabica* on *Phoma* and other pathogen infections.

Another topic to be investigated is the apparent difference between the mycobiomes of C. arabica present in different locations. As seen above, the mycobiomes found in other locations differ significantly from that presented here. While plants in Puerto Rico predominate *Pestalotia* and *Botryosphaeria* [9], those in southern Mexico and southern Brazil predominate *Colletotrichum* [12,13]. In our samples of C. arabica, these fungi were not detected, and *Cryptococcus* and *Cladosporium* predominate. The reason for these differences may be differences in the edaphoclimatic characteristics in these different locations (Puerto Rico and southern Mexico are wetter than southwestern Brazil), as previously seen in forest species [38,39]. Another reason could be more methodological: our work was carried out with the use of indirect detection via ITS sequencing, while the other works performed direct detection via isolation, which can distort the real structure of the mycobiome. The culture-independent approach can be conducted in more localities to gain a better overview of the C. arabica mycobiome.

## Figures and Tables

**Figure 1 microorganisms-09-02296-f001:**
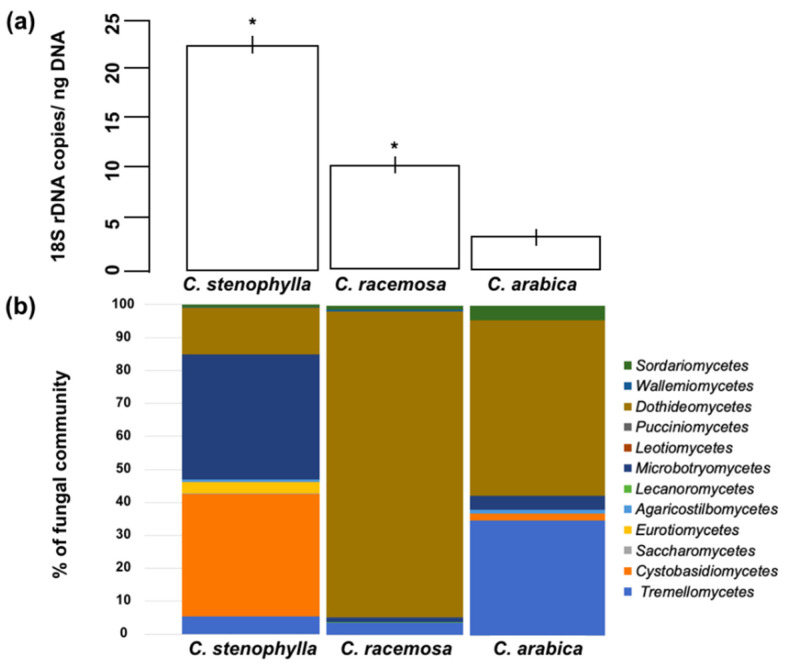
Fungal abundance and distribution in the phyllosphere of *Coffea*. (**a**) Total fungal abundance in the phyllosphere by the number of 18S rRNA copies per nanogram of total DNA. * Significant difference when comparing with *C. arabica* (*p* < 0.05). (**b**) Fungal distribution in the phyllosphere at the “class” level.

**Figure 2 microorganisms-09-02296-f002:**
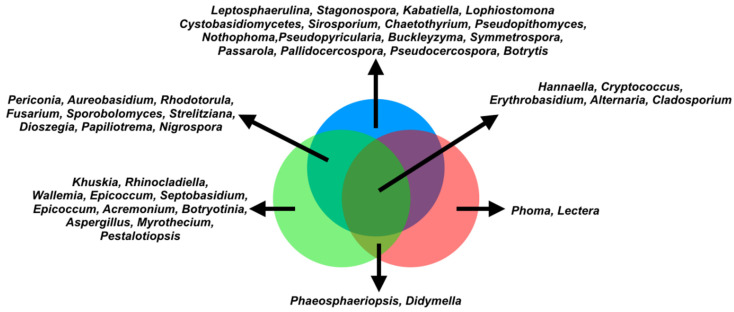
Venn diagram with differential distribution of genera. *C. stenophylla* (blue circle), *C. racemosa* (green circle), and *C. arabica* (red circle). Each arrow represents specific fungi for each coffee species. The central area represents the core mycobiome, fungi found in all three species. *C. racemosa* and *C. stenophylla* showed the highest number of species-specific fungi with 13 genera for *C. racemosa* and 17 genera for *C. stenophylla*, while *C. arabica* had only 2 specific genera: *Phoma* and *Lectera*. Only nine genera were present in *C. stenophylla* and *C. racemosa*, and only two genera were present in *C. racemosa* and *C. arabica* (Figure 2). No fungal genera were present in *C. stenophylla* and *C. arabica* (Figure 2).

**Figure 3 microorganisms-09-02296-f003:**
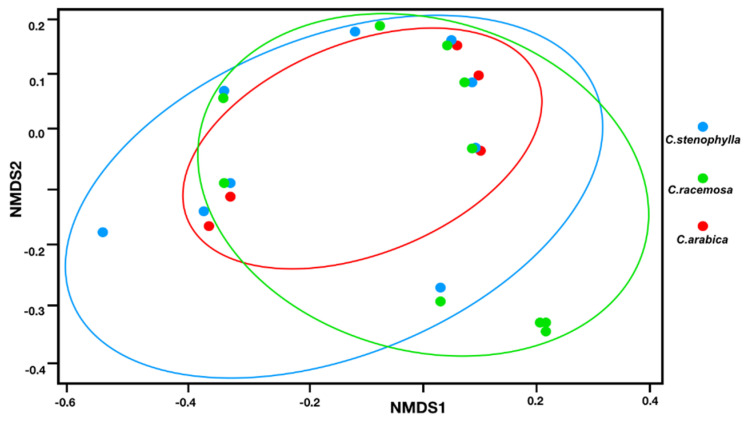
Beta diversity analysis using nonmetric multidimensional scaling (NMDS) plots based on Bray–Curtis dissimilarity of discriminating genera in the phyllosphere of *C. stenophylla* (blue line), *C. racemosa* (green line), and *C. arabica* (red line).

**Figure 4 microorganisms-09-02296-f004:**
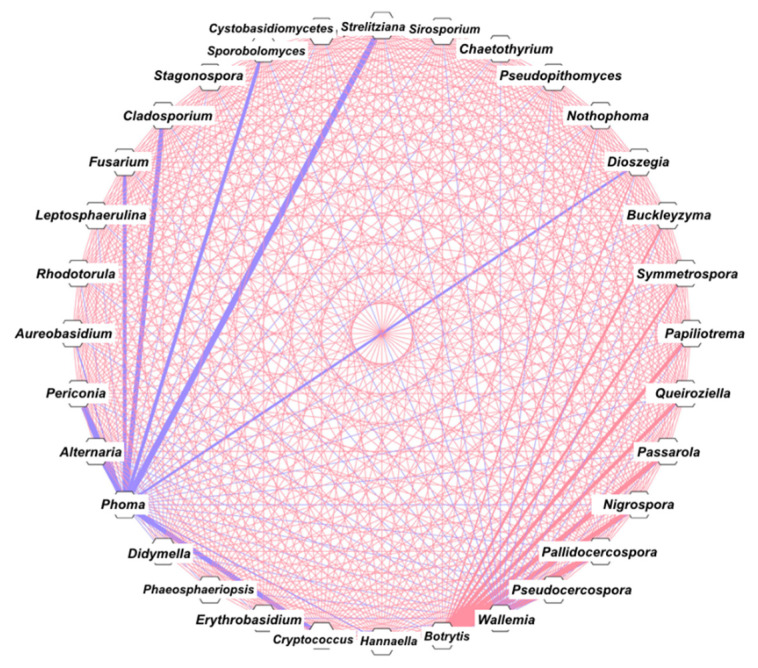
Network of mycobiome based on correlation analysis from taxonomic profiles. Each site corresponds to a genus. Cytoscape 3.7.1 was used for network construction. The thicker blue line represents strong negative interaction (Spearman’s rho < −0.8) and the thicker red line represents strong positive interaction (Spearman’s rho > 0.8).

**Table 1 microorganisms-09-02296-t001:** Concentration of caffeine and chlorogenic acid in three *Coffea* species measured by HPLC. Values are described in milligrams of the compound per gram of leaves.

	Caffeine (mg/g)	Chlorogenic Acid (mg/g)
*C. arabica*	7.4 ± 0.5	55 ± 3
*C. stenophylla*	0.26 ± 0.3	20 ± 2.5
*C. racemosa*	0.21 ± 0.1	23 ± 3.2

## Data Availability

All data presented in this study are available in the article.

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
