# Peer review of "Differences between the Leaf Mycobiome of Coffea arabica and Wild Coffee Species and Their Modulation by Caffeine/Chlorogenic Acid Content"

_microorganisms, 2021, doi:10.3390/microorganisms9112296_

Round 1
Reviewer 1 Report
The manuscript titled "Differences between the leaf mycobiome of Coffea arabica and wild coffee species" treats contents for Microorganisms Journal.
The introduction provide sufficient background and include all relevant references; the methods are adequately described; the results are clearly presented; the conclusions are supported by the results.
To verify some references (ref. 17, 18, 24..) abbreviation Journal omitted or not adequate..
Reviewer 2 Report
This is a very simple and straightforward study analyzing mycobiome. The manuscript is written well but would benefit from a further compare contrast of domesticated and wild strains. Below are some specific comments to improve the manuscript.
Abstract ïƒ C. arabica mycobiome is poor qualitatively and quantitatively compared to C. racemosa and C. stenophylla.
ïƒ Meaning is not clear
Introduction focuses more on coffee, distribution, and others. The introduction should more focus on microbes associated. In the second last paragraph of the introduction, the authors stated that the study is small. But there are 6 references, 6 is not small. Please focus on summarizing the finding of these six studies in terms of microbiota.
2.1 the statement is confusing. The plants were obtained from the store? Perhaps they were grown before leaf collection.
Provide details on how the interaction was analyzed in “2.5. Interaction networks between mycobiomes”?
Minor:
0,9% NaCl solution and where applicable, change to 0.9%
cording to Mazzafera et al [21] and Ramiro et al [22] specifications ïƒ Please include detail of (solvent, column, detector, etc) for identification.
Figure 2, perhaps, authors can use easily distinguishable colour
A table would be better for the data of caffeine and chlorogenic acids in 3.5
The authors used the word “secondary metabolite” in the manuscript and analyzed caffeine (CAF) and chlorogenic acid (CGA). When the word “secondary metabolite” is used, readers might get the impression that multiple compounds were analyzed. Yes CAF and CGA are secondary metabolites but as you have analyzed these two compounds only, it's better to use the compound names.
Author Response
Reviewer - This is a very simple and straightforward study analyzing mycobiome. The manuscript is written well but would benefit from a further compare contrast of domesticated and wild strains. Below are some specific comments to improve the manuscript.
Authors - A paragraph was written discussing this comparison. Thanks for the sugestion.
Reviewer - “C. arabica mycobiome is poor qualitatively and quantitatively compared to C. racemosa and C. stenophylla”. Meaning is not clear
Authors - That sentence has been withdrawn.
Reviewer - Introduction focuses more on coffee, distribution, and others. The introduction should more focus on microbes associated. In the second last paragraph of the introduction, the authors stated that the study is small. But there are 6 references, 6 is not small. Please focus on summarizing the finding of these six studies in terms of microbiota.
Authors - A paragraph was written commenting on what was asked, but focusing only on the fungi. Thanks for the suggestion.
Reviewer - 2.1 the statement is confusing. The plants were obtained from the store? Perhaps they were grown before leaf collection.
Authors - Leaves were detached from adult plants present in the germplasm bank. A change was made to make the sentence clearer.
Reviewer - Provide details on how the interaction was analyzed in “2.5. Interaction networks between mycobiomes”?
Authors – Done. Thanks for the suggestion.
Reviewer - 0,9% NaCl solution and where applicable, change to 0.9%
Authors – Done.
Reviewer - according to Mazzafera et al [21] and Ramiro et al [22] specifications à Please include detail of (solvent, column, detector, etc) for identification.
Authors – Done.
Reviewer - Figure 2, perhaps, authors can use easily distinguishable colour
Authors – Done.
Reviewer - A table would be better for the data of caffeine and chlorogenic acids in 3.5
Authors – Done.
Reviewer - The authors used the word “secondary metabolite” in the manuscript and analyzed caffeine (CAF) and chlorogenic acid (CGA). When the word “secondary metabolite” is used, readers might get the impression that multiple compounds were analyzed. Yes CAF and CGA are secondary metabolites but as you have analyzed these two compounds only, it's better to use the compound names.
Authors – Agree. Thanks for the suggestion.
Reviewer 3 Report
Reviewer #2
Manuscript Title: Differences between the leaf mycobiome of Coffea arabica and wild coffee species
Authors: Leandro Pio de Sousa et al., 2021
In this study, authors reported, the microbiome how associated with the coffee leaves particularly Coffea arabica and wild coffee species. Also, authors defined two major research problems, mainly, a) prospect microbes that can be used for biocontrol of diseases and pests; b) understand the influence of the host on the microbiome. Therefore, the work is very interesting and manuscript is well written and organized properly.
However, the manuscript should be improve the presentation particularly, typos and spell errors that need to correct it before publication.
Comments#
- 1 mention the year samples collected, 28th July 2020-1?
- 2: Typo mistake edit- 0.9% Nacl
- 2 mention the product/ instrument details (Name, product version, year, state, country) and check the information throughout text
- Methods: statistical information is missing. Author should include the details in the materials and methods section
- Figure 4 and 5 caption was very short and no details.
Briefly, provide the information in the captions
- Section 3.5, The data about (CAF) concentration was shown in the manuscript, but chlorogenic acid concentrations is not available in the text. What about the CGA concentration!
Author Response
Reviewer - mention the year samples collected, 28th July 2020-1?
Authors – 2021. The year was placed.
Reviewer - Typo mistake edit- 0.9% Nacl
Authors – Corrected. Thanks
Reviewer - mention the product/ instrument details (Name, product version, year, state, country) and check the information throughout text
Authors – Done.
Reviewer - Methods: statistical information is missing. Author should include the details in the materials and methods section
Authors – Done
Reviewer - Figure 4 and 5 caption was very short and no details.
Authors – Captions were better detailed. Thanks for the suggestion.
Briefly, provide the information in the captions
Section 3.5, The data about (CAF) concentration was shown in the manuscript, but chlorogenic acid concentrations is not available in the text. What about the CGA concentration!
Authors – A table has been placed for better understanding.
Round 2
Reviewer 2 Report
The authors did a great job revising the manuscript.